

# Long-term consumption of artificial sweeteners does not affect cardiovascular health and survival in rats

Satvinder K. Guru, Ying Li, Olga V. Savinova and Youhua Zhang

Department of Biomedical Sciences, New York Institute of Technology College of Osteopathic Medicine, Old Westbury, NY, United States of America

## ABSTRACT

**Background**. Recent epidemiological cohort studies have suggested that consumption of artificial sweeteners (AS) is associated with adverse cardiovascular events and mortality. However, these population association studies cannot establish a causal relationship. In this study we investigated the effect of long-term (1-year) consumption of AS (Equal and Splenda, two commonly used AS) on cardiovascular health and survival in rats.

**Methods**. Adult Sprague-Dawley rats (both sexes, 4–5 months old) were randomized into the following 3 groups: control ($n = 21$), AS Equal ($n = 21$) and Splenda ($n = 18$). In the AS groups, Equal or Splenda was added to the drinking water (2-packets/250 ml), while drinking water alone was used in the control rats. The treatment was administered for 12 months. Cardiovascular function and survival were monitored in all animals.

**Results**. It was found that rats in the AS groups consistently consumed more sweetened water than those in the control group. AS did not affect body weight, non-fasting blood cholesterol, triglycerides, blood pressure or pulse wave velocity. There were no significant differences in left ventricular wall thicknesses, chamber dimension, cardiac function or survival. AS did not affect heart rate or atrial effective refractory period. However, rats in both Equal and Splenda groups had prolonged PR intervals ($63 \pm 5$ms in Equal, $68 \pm 6$ ms in Splenda, vs $56 \pm 8$ ms in control, $p < 0.05$) and a tendency of increased atrial fibrillation inducibility.

**Conclusion**. Long-term consumption of AS does not affect cardiovascular structure, function or survival but may cause some electrophysiological abnormalities with prolonged PR intervals and a tendency of increased atrial fibrillation inducibility in rats.

Corresponding author
Youhua Zhang, yzhang49@nyit.edu

## INTRODUCTION

Sugar intake is associated with weight gain, type 2 diabetes mellitus, coronary heart disease, stroke and total mortality (*Malik et al., 2019*). Sugar-sweetened beverages (SSB) are the single largest source of added sugar in the US diet (*Hu & Malik, 2010*; *Drewnowski & Rehm, 2014*). Artificial sweeteners, with their intense sweetness and virtually no calories, are approved by the U.S. Food and Drug Administration (FDA) as sugar substitutes,

commonly used in foods and beverages. Currently, there are eight different artificial sweeteners approved by the FDA, including aspartame, sucralose, *etc* (*US Food and Drug Administration (FDA), 2022*). Artificial sweeteners are thought to be metabolically inert, contain only a few to no calories and do not raise blood sugar levels. The American Heart Association and American Diabetes Association have given a cautious nod to the use of artificial sweeteners in place of sugar to combat obesity, metabolic syndrome, and diabetes mellitus (*Gardner et al., 2012*). In 2018, the American Heart Association advised that short-term replacement of SSB with beverages containing low-calorie sweeteners, including artificially sweetened beverages (ASB), may be an effective and realistic approach to calorie reduction and weight loss in some adults (*Johnson et al., 2018*).

Since artificial sweeteners are approved by the FDA, they are generally considered to be safe for consumption. However, the safety of long-term use of artificial sweeteners is still a concern. For example, there is a concern about aspartame's carcinogenic potential, despite no evidence to conclusively prove it (*Mallikarjun & Sieburth, 2015*). There are also concerns that artificial sweeteners could affect glucose metabolism, body weight, diabetes, (*Pepino et al., 2013*; *Choudhary, 2018*) and gut microflora (*Abou-Donia et al., 2008*; *Pang, Goossens & Blaak, 2020*), *etc.* Recent epidemiological studies have shown an association between consumption of artificial sweeteners and adverse cardiovascular events—in particular ischemic stroke, coronary heart disease, and all-cause mortality (*Malik et al., 2019*; *Mossavar-Rahmani et al., 2019*; *Chazelas et al., 2020*). It is interesting that the association of ASBs intake and cardiovascular disease mortality was found mostly in women, but not in men (*Malik et al., 2019*). It should be noted that these epidemiological cohort studies were unavoidably affected by confounding and bias (*e.g.*, different types of ASBs and personal choice of the ASBs), thus could not establish a definite causal relationship. Only evidence from randomized controlled trials (the gold-standard in clinical research) can prove or disprove the cause–effect relationship.

Understandably, a long-term randomized controlled trial involving a large population is very difficult, if not impossible, to conduct. Currently there is no report of randomized controlled trials testing whether long-term consumption of artificial sweeteners causes adverse cardiovascular effects in humans. We hypothesized that long-term consumption of artificial sweeteners may adversely affect cardiovascular health and survival. Accordingly, this study was designed to test the above hypothesis and provide experimental evidence whether long-term (1-year, equivalent to 30 human years) consumption of artificial sweeteners (Equal and Splenda, two commonly used artificial sweeteners) causes adverse cardiovascular effects and survival in animals.

## MATERIALS & METHODS

### Study groups

There were two groups of adult (both sexes, 4–5 month-old, borne in-house) Sprague-Dawley rats. The first group of 33 rats was randomized into the artificial sweetener Equal (main ingredient: aspartame) group ($n = 21$) and control group ($n = 12$) in a 2:1 manner. The second group of 27 rats was randomized similarly in a 2:1 manner into the artificial
sweetener Splenda (main ingredient: sucralose) group ($n = 18$) and the control group ($n = 9$). As a result, the study consisted of the following three groups: (1) control group ($n = 21$), (2) artificial sweetener Equal (aspartame) group ($n = 21$), and (3) artificial sweetener Splenda (sucralose) group ($n = 18$). Either Equal or Splenda was added to the drinking water at a concentration of two packets per cup (8 Oz) of water (a high concentration for human consumption recommended by the manufacturers) in the respective groups. Note that both Equal and Splenda are commercial mixtures containing other ingredients besides the main ingredient aspartame or sucralose respectively. The calculated aspartame concentration was about 300 mg/L in Equal group, and the sucralose concentration was about 100 mg/L in Splenda group. Control animals drank only drinking water. Both Equal and Splenda were purchased from the market. The treatment was administered for 12 months. Body weight and water consumption were monitored before the treatment and at 2 weeks, 2 months, 4 months, 6 months, 9 months and 12 months after the treatment. At the end of the treatment, pulse wave velocity (an index of arterial stiffness) and cardiac echocardiogram were taken, together with invasive blood pressure, left ventricular hemodynamic measurement, and cardiac electrophysiological tests. Non-fasting blood samples were also taken at the end of the experiment for blood glucose and lipids analysis.

The use of the animals in this study was approved by the Institutional Animal Care and Use Committee (IACUC) at the New York Institute of Technology College of Osteopathic Medicine (IACUC approved protocol number: 2019-YZ-02) and was in accordance with the Guide for the Care and Use of Laboratory Animals. The animals were housed in our institutional animal care facility, fed with standard rat chow (Laboratory Rodent Diet-5001; LabDiet, St. Louis, MO, USA, with calories from protein 28.903%, fat 13.606% and carbohydrates 57.491%), kept on a 12-hour light/dark cycle with food and water available *ad libitum*.

## Pulse wave velocity measurement

As an index of arterial stiffness, pulse wave velocity (PWV) was measured at the end of the 12-month treatment. Rats were anesthetized with isoflurane (maintained at 1.5%) and PWV was measured utilizing an ECG-gated kilohertz visualization (EKV) on a VisualSonics Vevo 3100 platform equipped with a 40-MHz transducer (Fujifilm, Toronto, Canada). Time-resolved and averaged ultrasound data was collected from the longitudinal view of the left carotid artery at 3,000 frame/sec rate. Data were analyzed using VevoLab analysis software equipped with a VevoVasc module as previously reported (*Di Lascio et al., 2017*).

## Echocardiographic measurement

Echocardiographic measurement was taken just before the terminal experiments of hemodynamic study and electrophysiological test. VisualSonics Vevo 3100 platform was used coupled with an ultrasound transducer probe (25 MHz), as we have reported previously (*Zhang et al., 2014*). Echocardiograms were obtained from the short axis (at the papillary muscle level) and the long axis view of the left ventricle (LV) under isoflurane anesthesia (maintained at 1.5%). We determined LV wall thickness and chamber

dimensions in systole and diastole from short-axis views using two-dimensionally-targeted M-mode echocardiograms. The following parameters were measured: LV anterior wall thickness in end-systole and end-diastole, LV systolic and diastolic internal diameters, LV posterior wall thickness in end-systole and end-diastole, and LV fractional shortening. The left atrial (LA) diameter was determined at the aortic valve level from the parasternal long-axis view.

## Invasive blood pressure and left ventricular hemodynamic measurement

After echocardiographic measurements and just prior to cardiac electrophysiology and atrial fibrillation (AF) inducibility test, each rat had a 1.9F Scisense pressure catheter (Transonic Scisense, London, Ontario, Canada) inserted into the LV chamber through the right carotid artery, as previously described (*Zhang et al., 2013*). Placement of the catheter in the LV was verified by the LV pressure curve displayed in the data acquisition system (LabScribe). A 20-minute period was given for stabilization prior to the hemodynamic data collection. Based on the acquired LV pressure recording, the following LV hemodynamic data were calculated: LV systolic pressure (LVSP), LV end-diastolic pressure (LVEDP), positive change in LV pressure over time (LV +dP/dt), negative change in LV pressure over time (LV -dP/dt), and Tau (left ventricular relaxation time constant).

After completing the LV pressure recording, blood pressure was recorded by withdrawing the catheter from the LV to the ascending aorta.

## Cardiac electrophysiology and atrial fibrillation inducibility test

*In vivo* cardiac electrophysiology and atrial fibrillation (AF) inducibility tests were performed in all animals immediately after LV hemodynamic measurements, as previously described (*Zhang et al., 2013*; *Delfiner et al., 2018*). Rats were placed at supine position under isoflurane anesthesia (maintained at 1.5%). An octopolar Millar electrophysiology catheter (1.6F, EPR-802, Millar Instruments, In., Houston, Texas) was advanced into the right atrium (RA) through right jugular vein. The catheter has eight poles with three pairs of electrodes recording RA electrocardiograms and one pair for pacing. Standard ECG lead II and three RA ECG were recorded using a PowerLab data acquisition system (ADInstruments, Colorado Springs, CO).

Atrial effective refractory period (ERP) was determined using standard S1S2 pacing protocol (basic cycle length at 150ms, at x3 the atrial threshold). Atrial ERP was defined as the longest coupling intervals that did not capture the atria. The ERP was determined first by shortening of S2 interval in a 5-ms decrement and then by a 1-ms step. AF inducibility was tested with burst pacing containing 200 impulses at 100 Hz. Each rat received burst pacing five times and the subsequent AF duration after each burst pacing was documented. AF was defined as rapid irregular atrial activations with varying electrocardiographic morphology lasting $\geq 0.5$ s, as reported previously (*Zhang et al., 2013*). AF duration was calculated based on the average duration of AF over five burst pacing trials.

## Blood glucose and lipids assay

A Rat Glucose Assay kit (catalog # 81693) was used to quantify levels of glucose in non-fasted plasma according to the manufacturer's recommended protocol (Crystal Chem, Elk Grove Village, IL, USA).

Total cholesterol, triglycerides, and HDL-cholesterol were determined using specific reagents obtained from Pointe Scientific (Canton, MI, USA) in plasma collected before the animals were euthanized. Samples were analyzed using corresponding manual protocols in a 96-well format using a plate reader (BioTek Instruments, Winooski, VT, USA). Non-HDL cholesterol was calculated by subtracting HDL cholesterol from total cholesterol.

## Statistical analysis

Data are expressed as mean $\pm$ SD where appropriate. Normally distributed data such as body weight and cardiac functional parameters among the three groups were first evaluated using one-way analysis of variance (ANOVA) followed by post hoc analysis with Bonferroni correction for multiple comparisons if needed. Chi-square test was used to compare the AF incidence. Because the AF duration data were not normally distributed, they are expressed as median, first and third quartile (Q1–Q3) values. A nonparametric Kruskal–Wallis test followed by Mann–Whitney U tests were used to compare the AF duration data. Kaplan–Meier survival analysis with Log-rank (Mantel-Cox) test was used to compare the survival curves among the three groups. Statistical analysis was performed using GraphPad Prism 8.2.1 software. A $p < 0.05$ was accepted as statistically significant.

# RESULTS

## Body weight, water consumption, and survival

There were no significant differences in body weights among the three groups at baseline (before the treatment) and during the 1-year treatment in both female and male rats (Figs. 1A and 1B. Note that the body weight of males was heavier than females and both male and female rats gained weight as they grew).

After administration of artificial sweeteners, rats in both Equal and Splenda groups consistently consumed more sweetened water than those in the control group (Fig. 1C), indicating that the animals liked drinking sweetened water. Based on the amount of water consumed, the calculated aspartame intake ranged from 60–78 mg/kg/d, and sucralose intake ranged from 18–22 mg/kg/d in rats with respective treatments during the 1-year experiment. Both sweeteners exceeded the FDA acceptable daily intake (ADI, aspartame 50 mg/kg/d, sucralose 5 mg/kg/d) for human consumption (*Food & Drug Administration , FDA*). For reference, the daily amounts of aspartame and sucralose intake were equivalent to 20–50 (12-oz)-cans of diet soda (the aspartame and sucralose contents in diet soda were from *Wikipedia, 2022*) per day for a 60-kg (132-pound) person.

At the end of the experiment, 16/21 rats in the control group, 15/21 rats in the Equal group and 10/18 rats in the Splenda group survived. The survival was not significantly different among the three groups (Fig. 1D), indicating that the treatments did not affect survival. It should be pointed out that all rats that died were males except 1 female rat in the Equal group. Thus, the overall survival rate at the end of the 12-month experiment

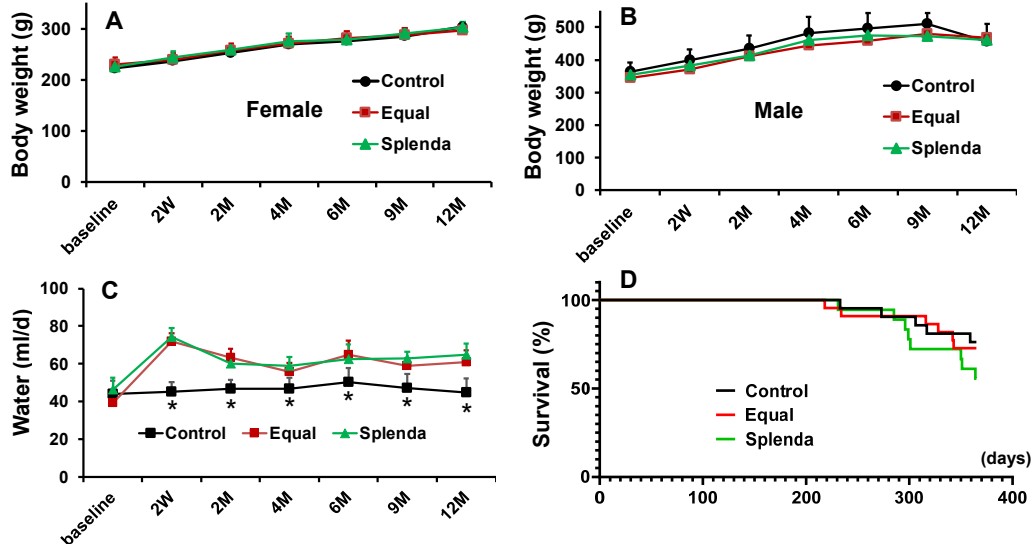

**Figure 1  Body weight, water consumption and survival.** (A) Body weight in female rats. (B) Body weight in male rats. (C) Water consumption. (D) Survival. Note that data are presented based on treatments with inclusion of both sexes, except as specified in A and B. *$p < 0.05$ *versus* both Equal and Splenda groups.

was higher in females (29/30 rats or 96.7% in females *versus* 12/30 rats or 40% in males, $p < 0.001$), indicating that female rats lived longer than males, as expected.

## Blood pressure and pulse wave velocity

Both systolic (139 ± 26 mmHg in the control, 142 ± 10 mmHg in the Equal, and 146 ± 17 mmHg in the Splenda group, ANOVA test, $F = 0.4751$, $P = 0.6256$) and diastolic (90 ± 20 mmHg in the control, 90 ± 8 mmHg in the Equal, and 94 ± 15 mmHg in the Splenda group, ANOVA test, $F = 0.3483$, $P = 0.7082$) blood pressures were similar among the three groups (Fig. 2A). The pulse wave velocity was also similar among the three groups (2.72 ± 0.54 m/s in the control, 2.83 ± 0.45 m/s in the Equal, and 3.13 ± 0.76 m/s in the Splenda group, ANOVA test, $F = 1.386$, $P = 0.2636$, Fig. 2B).

## Echocardiographic parameters

The left ventricular wall thickness, chamber dimension and function were similar among the three studied groups (Table 1).

## LV hemodynamic parameters

There were no differences in the LV hemodynamic parameters among the three groups except LVEDP. LVEDP was slightly but statistically higher in the Splenda group (Table 2). However, the value of LVEDP was still in the normal range.

## Electrophysiology and AF inducibility

The PR interval (AV conduction time) was significantly increased in both Equal (63 ± 5 ms) and Splenda (68 ± 6 ms) groups compared with that in the control (56 ± 8 ms) animals

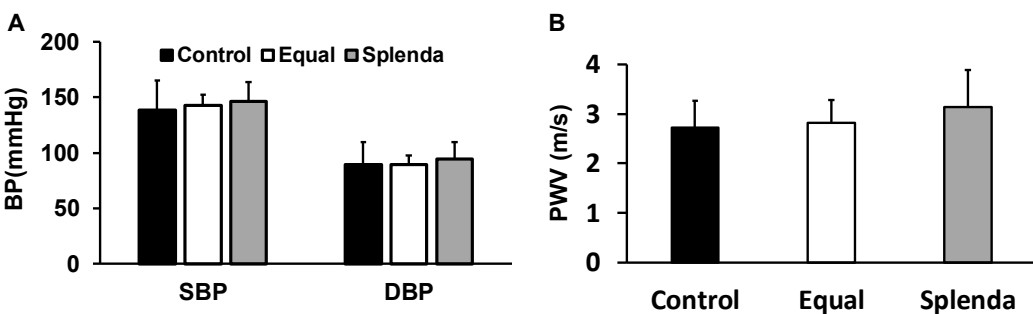

**Figure 2** **Blood pressure (BP) and pulse wave velocity (PWV).** (A) Systolic (SBP) and diastolic (DBP) blood pressure. (B) PWV. No statistical differences were found in all parameters.

**Table 1** **Echocardiographic measurements.**

|  | HR (bpm) | LVAWd (mm) | LVAWs (mm) | LVDd (mm) | LVDs (mm) | LVPWd (mm) | LVPWs (mm) | FS (%) | LA (mm) |
|---|---|---|---|---|---|---|---|---|---|
| Control | 271 ± 40 | 1.6 ± 0.2 | 2.7 ± 0.3 | 7.3 ± 0.9 | 3.7 ± 0.9 | 1.6 ± 0.2 | 2.8 ± 0.3 | 49.6 ± 7.3 | 3.8 ± 0.9 |
| Equal | 261 ± 20 | 1.6 ± 0.2 | 2.6 ± 0.2 | 6.9 ± 0.7 | 3.5 ± 0.7 | 1.6 ± 0.2 | 2.7 ± 0.2 | 49.6 ± 6.1 | 4.0 ± 0.7 |
| Splenda | 260 ± 31 | 1.5 ± 0.4 | 2.5 ± 0.5 | 7.7 ± 0.9 | 4.1 ± 0.9 | 1.5 ± 0.4 | 2.6 ± 0.5 | 46.6 ± 6.3 | 4.2 ± 0.6 |

**Notes.**

HR, heart rate; LVAWd, left ventricular (LV) anterior wall thickness in diastole; LVAWs, LV anterior wall thickness in systole; LVDd, Left ventricular diameter in diastole; LVDs, Left ventricular diameter in systole; LVPWd, LV posterior wall thickness in diastole; LVPWs, LV posterior wall thickness in systole; FS, fractional shortening; LA, left atrial diameter.

$P > 0.05$ for all parameters.

**Table 2** **LV hemodynamic measurements.**

|  | LVSP (mmHg) | LVEDP (mmHg) | +dp/dt (mmHg/s) | −dt/dt (mmHg/s) | Tau (ms) |
|---|---|---|---|---|---|
| Control | 135 ± 25 | 2.5 ± 2.2 | 8059 ± 2028 | −6864 ± 1991 | 13.7 ± 2.7 |
| Equal | 143 ± 11 | 3.2 ± 1.9 | 8056 ± 888 | −7322 ± 814 | 13.1 ± 1.6 |
| Splenda | 146 ± 19 | 6.1 ± 2.9* | 7804 ± 1005 | −6817 ± 579 | 14.5 ± 1.4 |

**Notes.**

LVSP, left ventricular systolic pressure; LVEDP, LV end-diastolic pressure; +dp/dt, maximal positive change in pressure over time; −dp/dt, maximal negative change in pressure over time; Tau, Left ventricular relaxation time constant.

*$P < 0.05$ versus Control.

(ANOVA test, $F = 10.84$, $P = 0.0002$). Atrial effective refractory period (ERP, 43 ± 13 ms in the control, 37 ± 6 ms in the Equal, and 39 ± 7 ms in the Splenda group, ANOVA test, $F = 1.619$, $P = 0.2114$) was not significantly different among the three groups (Fig. 3).

Figures 4A, 4B shows the original ECG (lead II) and right atrial electrocardiograms (RA) traces during AF inducibility test. Figure 4A shows an example that burst pacing did not induce AF, while an example that AF was induced immediately after the burst pacing is shown in Fig. 4B. The AF inducibility was higher in both Equal and Splenda groups compared with that in the control group, but the difference did not reach statistical significance (Chi-square test, $P = 0.1092$, Fig. 4C). The AF duration similarly showed

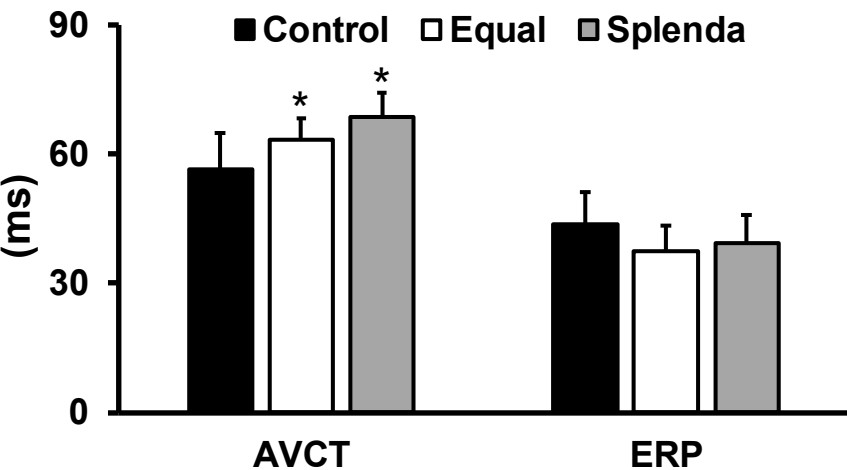

**Figure 3** Atriventricular conduction time (AVCT or PR interval) and atrial effective refractory period (ERP). $*p < 0.05$ *versus* control group.

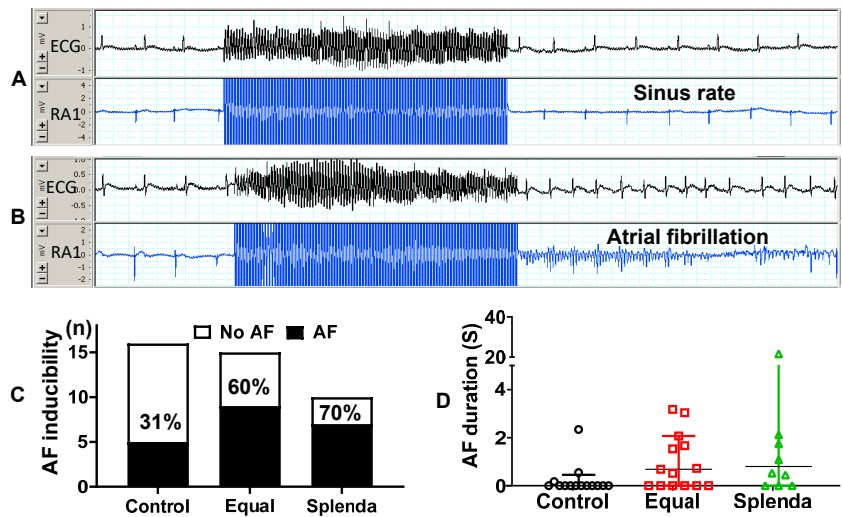

**Figure 4** Atrial fibrillation inducibility and duration. (A) Original ECG traces show an example that burst pacing did not induce AF. (B) An example that AF was induced immediately after the burst pacing. (C) The AF inducibility was higher in both Equal and Splenda groups compared with that in the control group, but the difference did not reach statistical significance (Chi-square test, $P = 0.1092$). (D) The AF duration similarly showed an increased tendency in both Equal and Splenda groups, but the difference was not statistically significant (Kruskal–Wallis test, $P = 0.1795$).

an increased tendency in both Equal and Splenda groups, but the difference was not statistically significant (Kruskal–Wallis test, $P = 0.1795$, Fig. 4D).

## Blood lipids and glucose level

There were no significant differences in blood triglycerides ($142 \pm 99$ mg/dl in control, $123 \pm 95$ mg/dl in Equal, and $152 \pm 110$ mg/dl in Splenda group, AVOVA test, $F = 0.2811$,

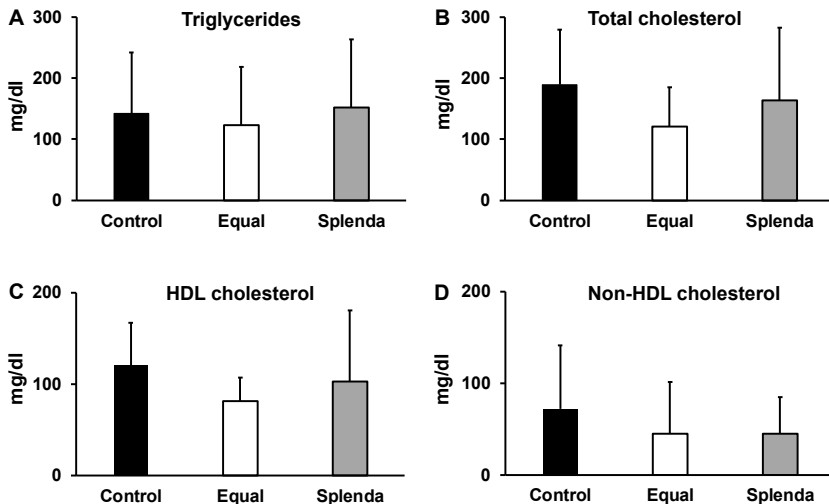

**Figure 5** **Blood lipids levels.** (A) Triglycerides. (B) Total cholesterol. (C) HDL cholesterol. (D) Non-HDL cholesterol. No statistical differences were found in all parameters.

$P = 0.7565$), total cholesterol ($189 \pm 89$ mg/dl in control, $121 \pm 63$ mg/dl in Equal, and $163 \pm 118$ mg/dl in Splenda group, AVOVA test, $F = 2.069$, $P = 0.1403$), HDL-cholesterol ($120 \pm 46$ mg/dl in control, $81 \pm 25$ mg/dl in Equal, and $102 \pm 77$ mg/dl in Splenda group, AVOVA test, $F = 2.446$, $P = 0.1005$), or non-HDL cholesterol ($71 \pm 69$ mg/dl in control, $45 \pm 56$ mg/dl in Equal, and $45 \pm 39$ mg/dl in Splenda, AVOVA test, $F = 1.011$, $P = 0.3736$) levels among the three groups at the end of 1-year treatment (Fig. 5).

The non-fasting blood glucose level was similar in the control group ($310 \pm 72$ mg/dl) and the Equal group ($308 \pm 82$ mg/dl, ANOVA test, $F = 4.418$, $P = 0.0188$, control *vs* Equal $P > 0.05$), but slightly lower in the Splenda group ($230 \pm 57$ mg/dl, $P < 0.05$ *versus* Control).

# DISCUSSION

## Major findings

In this study we administered artificial sweeteners in drinking water to avoid other potential variables, such as soda, caffeine and other active substances existed in various diet drinks. This design allowed us to directly compare consumption of artificial sweeteners with those without artificial sweeteners. Our results indicate that rats liked to drink artificially sweetened water. The daily water consumption was consistently higher in both Equal and Splenda groups than that in the control group (Fig. 1C). The long-term (1-year) consumption of artificial sweeteners did not significantly affect body weight, blood lipids, blood pressure, arterial stiffness (reflected in PWV), left ventricular wall thickness, chamber dimension, or cardiac function. Artificial sweeteners did not affect heart rate or atrial effective refractory period. However, the artificial sweeteners caused some electrophysiological abnormalities with a prolonged PR interval and a tendency of

increased atrial fibrillation (AF) inducibility in rats. The survival was not significantly affected by the 1-year treatments.

## Artificial sweeteners and adverse cardiovascular health

Artificial sweeteners are generally recognized as safe to pass the FDA approval. However, their health concerns have not been completely cleared. A report indicated that drinking diet beverages with artificial sweeteners was associated with an increased risk of heart attack and stroke (*Mossavar-Rahmani et al., 2019*). The study was conducted in a cohort of postmenopausal US women from the Women's Health Initiative Study, a multicenter longitudinal study of the health of 93,676 postmenopausal women. It was found that women who consumed an average of ≥2 ASB per day (≥24 ounces/day) had an elevated risk of stroke, coronary heart disease, and all-cause mortality compared with those who consumed <1 ASB per week. In a separate study (*Malik et al., 2019*), associations between consumption of ASB and risks of total and cause-specific mortality were examined among 37,716 men from the Health Professional's Follow-up study and 80,647 women from the Nurses' Health study who were free from chronic diseases at baseline. After adjusting for major diet and lifestyle factors, consumption of ASB was associated with total and cardiovascular disease mortality in the highest intake category only. In cohort-specific analysis, ASB was associated with mortality in women from the Nurses' Health Study but not in men from Health Professionals Follow-up Study (*Malik et al., 2019*), suggesting a sex difference in ASB induced adverse cardiovascular effects. Along this line, sexes and species differences in artificial sweeteners' effects have also been reported (*Marshall et al., 2017*; *Sclafani et al., 2010*).

A recent prospective cohort study found that both artificially sweetened beverages and sugary drinks were associated with increased cardiovascular disease risk (*Chazelas et al., 2020*). These findings certainly raised concerns about the adverse effects of artificial sweeteners on cardiovascular health (*Gardener & Elkind, 2019*). However, these findings are consistent with some, but not all, previous reports (*Gardener et al., 2012*; *Pase et al., 2017*; *De Koning et al., 2012*). In addition, these observational cohort studies are prone to confounding and bias. For example, participants consumed various diet beverages, which contain different kinds of artificial sweeteners and other active substances, making it difficult to isolate the effect of artificial sweeteners (needless to say a particular artificial sweetener). In addition, participants might have other biases in choosing ASBs. Thus, these cohort studies cannot establish a causal effect. Data from randomized controlled trials are needed to prove or disprove the causal effect and currently there is no randomized controlled trial available examining the long-term artificial sweetener consumption on cardiovascular health in humans. There are a few randomized controlled trials available that reported the effect of artificial sweeteners on body weight, blood lipids, glucose and blood pressure, (*Knopp, Brandt & Arky, 1976*; *Frey, 1976*; *Nichol, Holle & An, 2018*) but with few participants and short duration, the confidence in the reported results is limited (*Toews et al., 2019*).

To our knowledge this is the first study specifically designed to examine the effect of long-term consumption of artificial sweeteners on cardiovascular health in an

animal model. In this study, artificial sweeteners were administered in drinking water, which avoided many active substances and potential confounding existed in ASBs. We found that long-term consumption of artificial sweeteners did not affect body weight, blood pressure, arterial stiffness, cardiac function, chamber dimension, and mortality. However, artificial sweeteners caused some electrophysiological abnormalities, such as a prolonged PR interval and a tendency for increased AF inducibility in rats. AF is the most common, clinically significant arrhythmia and the number of AF patients is expected to increase worldwide (*January et al., 2019*). AF is a known risk factor for increased stroke and cardiovascular mortality (*January et al., 2019*). If artificial sweeteners could cause electrophysiological alterations leading to AF, then long-term consumption of artificial sweeteners would contribute to increased arrhythmias and stroke in patients, as suggested in the cohort studies (*Malik et al., 2019*). However, the increase of AF was not statistically significant in this study. This could be due to a relatively small sample size at the end of the experiment owing to high mortality in male rats. Thus, we think that the increased AF tendency found in the treated rats is important and deserves further investigation both experimentally and clinically in patients.

Our results indicate that long-term consumption of artificial sweeteners did not significantly affect left ventricular hemodynamics, except that Splenda caused a slight but statistically significant elevation in LVEDP. Despite the fact that the LVEDP value was still in the normal range, it may indicate early signs of impaired LV relaxation. *Risdon et al. (2020)* have reported that consumption of a mixture of sucralose/acesulfame potassium impaired vascular endothelial function in adult rats. These results are interesting and if confirmed, they could help to explain the increased cardio-metabolic risk reported by the epidemiological cohort studies (*Malik et al., 2019*; *Mossavar-Rahmani et al., 2019*; *Chazelas et al., 2020*).

## Comparison with previous reports

In a double-blind randomized study, it was found that consumption of aspartame for 13-weeks did not have a meaningful effect on body weight, plasma triglyceride or cholesterol levels in young adults (*Knopp, Brandt & Arky, 1976*). Our results revealed that 1-year consumption of artificial sweeteners did not affect body weight or blood lipid levels in rats, consistent with the report conducted in young adults. Similarly, in another 13-week, double-blind study conducted in apparently healthy children and adolescents, consumption of aspartame did not have clinically significant differences in laboratory parameters compared with those taking sucrose (*Frey, 1976*).

A meta-analysis of randomized controlled trials found that consumption of nonnutritive sweeteners did not increase blood glucose levels (*Nichol, Holle & An, 2018*). The glycemic impact of nonnutritive sweeteners consumption did not differ by the types of nonnutritive sweeteners but to some extent varied by participants' age, body weight, and diabetic status. A recent study in humans suggested that consuming sucralose-sweetened beverages with a carbohydrate (maltodextrin) impaired insulin sensitivity, but consuming either sucralose or carbohydrate alone did not affect insulin sensitivity (*Dalenberg et al., 2020*). Our data indicate that long-term consumption of artificial sweeteners in rats did not increase blood

glucose levels, compared with controls. In the current study blood glucose levels were similar in the control and the Equal groups, but slightly lower in the Splenda group, suggesting that long-term consumption of different types of artificial sweeteners may affect blood glucose levels differently.

It was reported that acute intraperitoneal injection of aspartame could decrease blood pressure in spontaneously hypertensive rats (*Kiritsy & Maher, 1986*). However, in the current study, we found that long-term consumption of artificial sweeteners (including Equal, which contains aspartame) did not affect blood pressure levels or arterial stiffness. These discrepancies could be caused by the dosages, route of administration, duration of the treatment or the animal conditions (spontaneously hypertensive rats *versus* normal control rats). As mentioned already, there is a report that consumption of a mixture of sucralose/acesulfame potassium impaired vascular endothelial function in rats (*Risdon et al., 2020*). However, in this study we did not observe significant changes in blood pressure and arterial stiffness in the Splenda and Equal treated animals.

A systematic review and meta-analysis summarized current evidence of artificial sweeteners on body weight (*Azad et al., 2017*). There were only seven randomized controlled trials (the gold standard in clinical research), involving 1003 participants, with a median follow-up of 6 months. There were 30 cohort studies (405 907 participants; median follow-up 10 years). In the included randomized controlled trials, nonnutritive sweeteners had no significant effect on body mass index. In the included cohort studies, consumption of nonnutritive sweeteners was associated with a modest increase in body mass index. The data from the randomized controlled trials did not show consistent effects of nonnutritive sweeteners on other measures of body composition. In the cohort studies, consumption of nonnutritive sweeteners was associated with increases in weight, waist circumference, and a higher incidence of obesity, hypertension, metabolic syndrome, type 2 diabetes and cardiovascular events (*Azad et al., 2017*), indicating a discrepancy in findings from randomized controlled trials and cohort studies. Another meta-analysis of randomized controlled trials indicated that substituting low calorie sweeteners for their regular-calorie versions resulted in a modest weight loss and therefore may be a useful dietary tool to improve compliance with weight loss or weight maintenance plans (*Miller & Perez, 2014*). There is also a report suggesting that different low-calorie sweeteners may have distinct effect on body weight (*Higgins & Mattes, 2019*). Our results indicate that long-term consumption of artificial sweeteners did not affect body weights in rats.

## Study limitations

During the 1-year experiment, the status of the animals was checked daily and recorded by our animal care facility personnel. The date of animal death (when it happened) was recorded. However, due to the COVID-19 lockdown in the early part of 2020 (when most deaths occurred), campus access was limited and autopsy for the dead animals was not performed. Thus, the exact cause of death for the animals was not known. Nevertheless, the survival of the rats in the artificial sweetener groups was not significantly different from that in the control group.

Blood glucose level was measured from non-fasted plasma samples taken at the end of the terminal experiments and the values in all groups (including the controls) were higher than typically seen in non-fasted rats. It is reported that surgical stress and anesthesia can elevate blood glucose level (*Duggan, Carlson & Umpierrez, 2017*). The extensive surgical procedures at the terminal experiments (hemodynamic measurements, electrophysiological tests, and open chest blood draw) might contribute to the elevated blood glucose levels in these non-diabetic rats.

Our data indicate that artificial sweeteners caused some electrophysiological abnormalities, such as a prolonged PR interval and a tendency for increased AF inducibility in rats. However, our data do not permit to delineate the mechanism(s) by which the artificial sweeteners caused these electrophysiological abnormalities.

Finally, due to the nature of an animal study, the findings from this study cannot be directly extrapolated to humans. Thus, human studies are still needed to verify these findings.

## CONCLUSION

Contrary to the hypothesis that long-term consumption of artificial sweeteners may adversely affect cardiovascular health and survival, consumption of artificial sweeteners did not affect body weight, blood lipids, blood pressure, cardiovascular structure, function, or survival in rats. However, artificial sweeteners caused some electrophysiological abnormalities, such as a prolonged PR interval and a tendency for increased AF inducibility in rats. Considering that AF is a known risk factor for stroke and other adverse cardiovascular events, further studies are warranted to clarify this issue and to determine whether artificial sweeteners lead to similar electrophysiological alternations in humans.

## ACKNOWLEDGEMENTS

The authors would like to thank Jeanne Quidore-Jermann, AAS, Senior Animal Care Technician, and Sandra Kahler at the New York Institute of Technology College of Osteopathic Medicine animal care facility for their expert care, monitoring, and administration of the artificial sweeteners to the animals during the experiment.

### Funding
The authors received no funding for this work.

### Competing Interests
The authors declare there are no competing interests.

### Author Contributions
- Satvinder K. Guru and Ying Li performed the experiments, analyzed the data, prepared figures and/or tables, and approved the final draft.

- Olga V. Savinova performed the experiments, analyzed the data, prepared figures and/or tables, authored or reviewed drafts of the paper, and approved the final draft.
- Youhua Zhang conceived and designed the experiments, performed the experiments, analyzed the data, prepared figures and/or tables, authored or reviewed drafts of the paper, and approved the final draft.

## Animal Ethics

The following information was supplied relating to ethical approvals (i.e., approving body and any reference numbers):

The Institutional Animal Care and Use Committee (IACUC) at the New York Institute of Technology College of Osteopathic Medicine approved this research (2019-YZ-02).

## Data Availability

The raw data are available in the Supplementary Files.

## Supplemental Information

Supplemental information for this article can be found online at http://dx.doi.org/10.7717/peerj.13071#supplemental-information.

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
