# Peer review of "Long-term consumption of artificial sweeteners does not affect cardiovascular health and survival in rats"

_PeerJ, doi:10.7717/peerj.13071_

## Round 0.1 · original submission · Minor Revisions

Dear Dr. Zhang,

Your manuscript entitled " Long-term consumption of artificial sweeteners does not affect cardiovascular health and survival in rats" which you submitted to PeerJ, has been reviewed by the editor and 3 experts in the field.

The reviewers are in general favorable and suggest that, subject to minor revisions, your paper could be suitable for publication. Please give these points your careful attention as the revised manuscript will undergo a second round of review by the same reviewers.

I hope that you will be prepared to make the necessary amendments and submit a revised manuscript accompanied by a statement of how you have responded to the reviewer’s comments, particularly those concerning the issue about the dosages of sweeteners, statistical analysis, and the electrophysiological findings.

Sincerely yours,
Stefano Menini

Reviewer 1 ·

Basic reporting

line 180-181: should be referenced properly.

Experimental design

no comment

Validity of the findings

no comment

Additional comments

The topic is interesting and the data unique. The authors present interesting data about the Long-term consumption of Artificial Sweeteners and how it does not affect cardiovascular health and survival in Rats especially with the current pandemic that affected research I find that it is a very interesting study with great results.
The manuscript is well written, has important message, and should be of great interest to the readers.

Reviewer 2 ·

Basic reporting

This is a straightforward paper. However, much of the scientific literature on artificial sweeteners has not been included.

Experimental design

The word "batch" should be changed to "group" throughout the manuscript. It is troublesome that fluid intake was not measured daily. This is necessary to make conclusions about dosages.

Validity of the findings

It is hard to determine the validity of findings due to the estimates of sweetened water intake as noted above. There were also issues of monitoring during the Covid crisis. However, the findings on electrophysiological alterations is a very important observation if correct. The prevalence of atrial fibrillation is increasing in the human population and should be addressed in the discussion.

Additional comments

A correlation between AFIB and artificial sweetener use would be helpful here.

Reviewer 3 ·

Basic reporting

The report is well structured; the language is clear and concise. The introduction provides a good context that leads to the research question. In line 58 the association of cardiovascular diseases and sex are considered as relevant background, but the possible influence of sex on cardiovascular diseases related to the sweetener used is not retaken. Considering the sample size of each group, it is possible to carry out a statistical analysis that considers this factor. Figure captions could explain better their content.

Experimental design

The study’s experimental design is ambiguous, it is not clear whether the authors seek to explore the differences between the sexes. For example, it is not clear if the comparison of weight and water consumption between groups was made by sex, that is, a comparison between groups for females and another for males, or if a comparison was made considering only the treatments (line 171).

Additionally, given that weight and water intake were collected repeatedly, it is convenient to perform an analysis that considers the repeated measures (for example, an ANOVA of repeated measures if the data meet the criteria of sphericity). If possible, I strongly suggest performing a statistical analysis that considers the effect of sex, treatment, and the repeated measures.
It is not clear whether the survival analysis was performed with both sexes or if each sex was analyzed separately. The results mention that survival was higher in females, but the authors do not mention the test used to compare the preference curves (line 186).
I suggest adding the mean of the groups ± SD, the values ​​of F (assuming that an ANOVA test was performed) and the value of p in blood pressure and pulse wave velocity results(line 190). In the same sense, the means for the AVCT and ERP measures, the value of the test used, and the adjusted p values ​​of the multiple comparisons (line 200) should be added, as well as the means of blood lipids and glucose levels, the test and p-values ​​(line 212). There is no reference chart for blood glucose values ​​(line 214).

Validity of the findings

The discussion sounds more like a review. I suggest that the authors compare how their findings fit in with the existing literature. A better discussion considering the differences between sexes could be written. I list some articles that report that the effect sweet foods is more pronounced in adult females (Valenstain, 1967, J Comp Physiol Psychol, 63 : 429-433; Sclafani et al., 2010, Chem Senses, 35: 433-443; Marshall et al., 2017, PLoS One, 12 [7]: e0180907).
Authors should mention that the sweeteners used in their research is a commercial mixture. Findings in humans suggest that the intake of beverages containing a mixture of sucralose and maltodextrin have different effects on the AUC of insulin related to the intake of beverages containing only sucralose or maltodextrin (Dalenberg et al., 2020, Cell Metab, 31: 493-502.e7). Therefore, I suggest the discussion should include literature with the effects of the specific sweeteners used in the study.
I consider that the most important result to discuss is that the prolonged use of splenda had an effect on left ventricular hemodynamics (line 225). In this regard, Risdon et al. (2021, Nutr Metab Cardiovasc Dis, 30: 843-846) have reported that the intake of a mixture of acesulfame and sucralose has an effect on the endothelial vascular function of adult rats. Certainly, the incorporation of related literature will help enrich the discussion.

Additional comments

It is interesting research with considerable data, but it lacks a concise approach to the interpretation of the results. A better discussion can give more validity to the work.

---

## Round 0.2 · accepted · Accept

Dear Dr. Zhang,

Thank you for submitting a revised version of your manuscript. I am pleased to inform you that your manuscript is accepted for publication in PeerJ in its current form.

Sincerely yours,
Stefano Menini